# Spatial distribution of stunting among breast feeding children in Sub-Sahara Africa

**Bekahegn Girma** [1]*, **Lealem Dibku Sasahu**[2], **Azizur Rahman**[3]

**1** School of Nursing and Midwifery, Asrat Woldeyes Health Science Campus, Debre Berhan University, Debre Berhan, Ethiopia, **2** Department of Natural Resource Management, College of Agriculture and Natural Resource, Dilla University, Dilla, Ethiopia, **3** School of Computing, Mathematics and Engineering, Charles Sturt University, Wagga Wagga, New South Wales, Australia

* Bekahagngirma@dbu.edu.et

## Abstract

### Background

Malnutrition is still a major global public health issue, especially in Sub-Saharan Africa (SSA), where millions of children suffer from stunting, a chronic form of the disease. In addition to limiting physical growth, stunting also impedes social and cognitive development, which frequently has long-term effects. Stunting is still incredibly common in SSA, with notable regional variations, despite international efforts to address hunger. Moreover, no studies have been conducted to assess the spatial distribution of stunting at the SSA level. Therefore, to pinpoint high-burden areas and guide focused treatments; this study intends to investigate the spatial distribution of stunting among breastfeeding children in SSA.

### Methods

The demographic and Health Survey data from 31 SSA nations were used for this study. The investigation included a total of 174,586 breastfeeding children. Stunting clustering and geographic patterns were evaluated using Geographic Information Systems and spatial analytic methods such as Getis-Ord Gi* and Global Moran's I. Stunting prevalence in unsampled areas was predicted using spatial interpolation (Kriging techniques). For the participants, descriptive statistics were calculated.

### Result

Stunting prevalence in SSA varied from 17.63% to 53.68%, with Madagascar (39.43%), Burundi (42.40%), and the Democratic Republic of the Congo (53.68%) having the highest rates. Significant clustering of stunting was found by spatial analysis (Moran's I: 0.639, p < 0.001), with hotspots primarily located in Central and Eastern Africa, such as Ethiopia, and the Democratic Republic of the Congo. In Southern Africa, cold locations like Namibia and South Africa were found to have lower stunting rates.

**Data availability statement:** The data used in this study were originally accessed and freely downloaded from the Demographic and Health Surveys (DHS) Program for sub-Saharan African countries (https://dhsprogram.com/data) after obtaining the necessary authorization. However, following recent changes in U.S. foreign policy, these data are no longer accessible. Consequently, the dataset analyzed during the current study is available from the corresponding author upon reasonable request and solely for purposes related to the replication or verification of this study.

**Funding:** The author(s) received no specific funding for this work.

**Competing interests:** All authors declare no competing interests.

**Abbreviations:** DHS, demographic and health surveys; GIS, geographic information systems; SSA, sub-Sahara Africa; WHO, World Health Organization.

## Conclusion

The significant regional variability in the prevalence of stunting among nursing children in SSA was highlighted by this study. In order to address underlying variables including poverty, food insecurity, and limited access to healthcare, the findings highlighted the necessity of region-specific public health initiatives. To lessen the burden of stunting and its long-term effects, improved geospatial surveillance systems are crucial for locating high-risk locations and directing the distribution of resources.

## Background

Sufficient nourishment is essential for children's physical and mental growth [1,2]. A child's overall development depends on vaccinations, growth monitoring, care and support in addition to a healthy diet. An important measure of a nation's social and economic progress is the nutrition of its children [3,4].

However, malnutrition is a major public health concern despite the worldwide efforts to enhance nutrition [5], child malnutrition remains a serious health concern, contributing to about half of all deaths in children under five worldwide [4,6]. Stunting, wasting, underweight, and overweight are some of the manifestations of malnutrition that have been linked to major causes of sickness and disability which hinders economic progress in the majority of developing nations [7].

Childhood stunting, a kind of malnutrition, has been recognized as a major impediment to human growth that occurs when a child's height is below average for their age [8]. Stunting is not limited to childhood. It has an impact on cognitive and physical development, and these effects may last a person's entire lifetime [9].

Globally, 149 million children under the age of five were expected to be stunted (too short for age) in 2022 [10]. A systematic review conducted in low- and middle-income countries (LMICs) reported a stunting prevalence of 43.4% [11], highlighting the widespread nature of the issue in these regions. [10]. According to one study done in Africa, 41% of under-five children are stunted [12]. Another meta-analysis study also conducted in Africa identified two regions with the highest prevalence of stunting, East Africa and Central Africa [13].

Moreover, systematic and meta-analyses conducted in SSA reported a stunting prevalence of 35% [12,13]. Studies conducted in different African countries tried to show the spatial distribution of stunting at the national level [14–18]. However, no comprehensive summary currently exists that illustrates the spatial distribution of stunting among children at the Sub-Saharan Africa (SSA) level, particularly among those who are breastfed. This gap limits the ability of policymakers and public health practitioners to design geographically targeted interventions for vulnerable subgroups.

Low-income households, home births, filthy cooking fuel, mothers' lack of formal education, being male, feeding practices, environmental factors, and children's inadequate dairy product intake are identified risk factors for childhood stunting [12,19–21].

To prevent malnutrition and encourage healthy growth, optimal breastfeeding practices are essential. For the first six months of life, the World Health Organization

(WHO) advises exclusive breastfeeding. After that, the child should continue to be breastfed for at least two years, coupled with appropriate supplemental nutrition [22]. According to Victora et al [23], breast milk offers vital nutrients, boosts immunity, and lowers the risk of infections—all of which are major causes of stunting. However, suboptimal breastfeeding practices, influenced by cultural, social, and economic factors, remain prevalent in SSA, limiting their protective effect against stunting [24].

Due to geographical variations in food supply, healthcare access, sanitation, and sociocultural customs, the prevalence of stunting in SSA shows notable spatial heterogeneity. Designing focused interventions requires an understanding of these geographical variances. Spatial analytic techniques and Geographic Information Systems (GIS) offer strong instruments for determining high-risk regions, facilitating evidence-based resource allocation and policymaking [25].

The causes of stunting in SSA have been the subject of numerous studies and there are some studies conducted to assess the spatial distribution of stunting in selected countries [26–29]. However, little was known about how was occurs geographically in breastfeeding children at the continent level. Finding priority/hotspot areas and developing specialized interventions to alleviate the burden of stunting in SSA could be facilitated by looking into the geographic patterns of stunting. Therefore, this study sought to evaluate the regional distribution of stunting in SSA among infants who were fed breast milk.

## Methods

### Study setting and period

There are 48 countries in SSA. Nevertheless, DHS data was unavailable for six of these nations. Seven of the nations that had DHS data lacked recent data (after 2010). Furthermore, weight and height measurements were absent from the DHS data for two nations (Senegal and Angola). Last but not least, Zambia and Congo-Brazzaville lacked geographic data. Consequently, this study was carried out only in 31 SSA nations which found in SSA (Table 1).

### Population and sample

We used data from current DHS records for SSA countries. The study population was children in 31 SSA nations who were found in the chosen clusters and were breastfeeding during the data collection period. After records of children with missing information and data points with 0 degrees for both longitude and latitude were excluded a total of 174,586 children from15398 clusters were included in this study,

We used the DHS and geographic data to perform this spatial analysis study. The DHS employed stratified cluster sampling in two stages. Using probability sampling, clusters and enumeration areas were chosen in the initial stage. In the second phase, probability sampling was used to choose households within the chosen cluster. All under-five children in each chosen family had their anthropometry measured, breastfeeding during the survey period and mothers between the ages of 15 and 49 were questioned.

### Eligibility criteria

Children under five who were breastfeeding at the time of the survey, had clusters with longitude and latitude data, and had accurate anthropometric measurements (height-for-age Z-scores). Only survey clusters with geocoded data were included to make spatial analysis easier. The study excluded children who were not breastfeeding at the time of the survey, children from nations without geocoded data due to security or confidentiality issues, and children with missing or implausible anthropometric measurements.

### Variables and data sources

Stunting in children under five was the study's outcome variable, and it was derived from the most recent DHS data from the participating nations. If a child's height for age falls below −2 Standard Deviations (SD), they are deemed stunted. The

 

**Table 1. Eligible Sub-Saharan African countries for this Study, 2025.**

| Sub-regions of Sub-Saharan Africa | List of countries | Recent DHS Year (Post 2010) | Measurements (weight & height) | Geographic dataset | Status |
|---|---|---|---|---|---|
| Central Africa | Angola | 2015/16 | No | Yes | Not eligible |
| | Cameroon | 2018 | Yes | Yes | Eligible |
| | Central Africa Republic | No-recent data | -------- | -------- | Not eligible |
| | Chad | 2014/15 | Yes | Yes | Eligible |
| | Congo demographic republic (Brazzaville) | 2011/12 | Yes | Yes | Eligible |
| | Congo (Kinshas) | 2013/14 | yes | No | Not eligible |
| | Equatorial guinea | No data | --------- | ---- | Not eligible |
| | Gabon | 2019/21 | yes | yes | Eligible |
| | Sao Tome and Principe | No recent data | --------- | ------ | Not eligible |
| Southern Africa | Botswana | No recent data | --------- | ------- | Not eligible |
| | Lesotho | 2014 | yes | Yes | Eligible |
| | Namibia | 2013 | Yes | Yes | Eligible |
| | South Africa | 2016 | Yes | Yes | Eligible |
| | Eswatini (Swaziland) | No recent data | --------- | Yes | Not eligible |
| Western Africa | Benin | 2017/18 | yes | Yes | Eligible |
| | Burkina Faso | 2021 | Yes | Yes | Eligible |
| | Cape Verde | No recent data | --------- | ------ | Not eligible |
| | Cote divore | 2011/12 | Yes | Yes | Eligible |
| | Gambia | 2019/20 | Yes | Yes | Eligible |
| | Ghana | 2014 | Yes | Yes | Eligible |
| | Guine | 2018 | Yes | Yes | Eligible |
| | Gunie Bissau | No data | --------- | ----- | Not eligible |
| | Liberia | 2019/20 | Yes | Yes | Eligible |
| | Mali | 2018 | Yes | Yes | Eligible |
| | Mauritania | 2019/21 | Yes | Yes | Eligible |
| | Niger | 2012 | Yes | yes | Eligible |
| | Nigeria | 2018 | Yes | Yes | Eligible |
| | Senegal | 2023 | No | Yes | Not Eligible |
| | Sierra Leone | 2019 | Yes | Yes | Eligible |
| | Togo | 2013/14 | Yes | Yes | Eligible |
| East Africa | Burundi | 2016/17 | Yes | Yes | Eligible |
| | Comoros | 2012 | Yes | Yes | Eligible |
| | Eritrea | No recent data | --------- | ------ | Not eligible |
| | Ethiopia | 2016 | Yes | Yes | Eligible |
| | Kenya | 2022 | Yes | Yes | Eligible |
| | Madagascar | 2021 | Yes | Yes | Eligible |
| | Malawi | 2015/2016 | Yes | Yes | Eligible |
| | Mauritius | No data | --------- | ------ | Not eligible |
| | Mozambique | 2022 | Yes | Yes | Eligible |
| | Rwanda | 2019/20 | Yes | Yes | Eligible |
| | Somalia | No data | --------- | ------ | Not eligible |
| | Sudan | No-recent data | --------- | ------- | Not eligible |
| | South Sudan | No data | ------ | ------ | Not eligible |
| | Tanzania | 2022 | Yes | Yes | Eligible |
| | Uganda | 2016 | Yes | Yes | Eligible |
| | Zambia | 2018 | --------- | ------ | Not eligible |
| | Zimbabwe | 2015 | Yes | Yes | Eligible |
| | Seychelles | No data | --------- | -- | Not eligible |
| **Total eligible countries with DHS-datasets** | | | | | **31** |

DHS data is reflective of the entire country. The Global Administrative Areas (GADM), a free online database, provided the shape file for Africa's borders. Using ArcGIS version 10.7.1 software, the percentage of stunting was geo-referenced and connected to area-level variables.

## Data processing and analysis

Formal registrations and requests were made to access the DHS data on the DHS website. This research was conducted using the Kids Records (KR) datasets. The table displays descriptive statistics that were computed, such as the percentage of stunting in each nation. If the child's score was less than −2SD, the outcome variable was categorized as stunting.

To take into consideration varying response rates and sampling probabilities, the data were weighted. Weighted frequencies and percentages for variables at the community and individual levels were employed as descriptive measures to provide an overview of the research participants' characteristics.

The degree to which the prevalence of stunting is similar or different in adjacent geographic places was examined by spatial autocorrelation using ArcGIS 10.7.1 software. In order to ascertain if the frequency of stunting in Sub-Saharan Africa follows a clustered, scattered, or random spatial pattern, this study was compute Global Moran's I, its value ranges from −1 to +1.

The formula for Moran's I is

$$I = \frac{N}{W} \times \frac{\sum_{i=1}^{N} \sum_{j=1}^{N} \omega ij \left(xi - \bar{x}\right)\left(xj - \bar{x}\right)}{\sum_{i=1}^{N} \left(xi - \bar{x}\right)^2}$$

Where:

I = Moran's I statistic

N = Number of spatial units (e.g., districts, regions)

Xi = Value of the variable at location i

$\bar{x}$ = Mean of the variable across all locations

$\omega ij$ = Spatial weight between location i and j (from a spatial weights matrix)

W=$\sum_{i=1}^{N} \sum_{j=1}^{N} \omega ij$ = Sum of all spatial weights

If the test statistics are significant (P < 0.05), the observed pattern of stunting displays randomness, while if they are not, the Moran's I value approaches −1, which indicates dispersion, and +1, which indicates clustered. Furthermore, Local Moran's I pinpointed outliers (high-low or low-high) and certain clusters (high-high or low-low locations). By shedding light on regional differences in stunting, these analyses aided in identifying regions that continue to face public health issues.

The Getis-Ord Gi was used to identify areas of stunting with high and low prevalence. Gi* is used to identify and indicate hotspot and cold spot regions. Using information from surveyed clusters, spatial interpolation produced a continuous surface of values that projected the frequency of stunting in unsampled areas. To estimate stunting rates throughout Sub-Saharan Africa, this study used Kriging technique, which considered spatial autocorrelation. In areas with a high prevalence of stunting, the produced maps enabled targeted treatments showed spatial patterns, and identified data gaps.

## Ethical consideration

This study was conducted by the principles of the Declaration of Helsinki. The researchers obtained authorization to access and utilize the dataset from the DHS program. A publicly available DHS Data was used in this investigation. The DHS data are IRB-approved and fully de-identified. All participants gave their informed consent when the data was first being collected. By ensuring that no identifying information was present, the dataset was anonymzed, protecting the participants' privacy and confidentiality.

# Result

## Socio-demographic characteristics of the included participants

Among the included study participants, 50.38% were male. Seventy-four thousand two hundred thirty-seven (42.52%) of children were born from mothers who had no education. Lastly, 128,672 (73.70%) of the study population were rural residents (Table 2).

## Proportion of stunting in SSA countries

As illustrated in Table 3, countries with the lowest rates of stunting were Burkina Faso, Ghana, Kenya, and the Gambia at 17.63, 18.36, 18.53, and 20.12%, respectively. However, stunting rates were highest in Madagascar, Burundi, and the Congo Democratic Republic, at 39.43, 42.40, and 53.68%, respectively. Lastly, the prevalence of stunting in SSA varied highly, it ranged from 17.63 up to 53.68 percent.

## Spatial distribution of stunting in SSA

With a Moran's Index of 0.639, there was a strong positive spatial autocorrelation. This number suggested that stunting showed geographical clustering rather than being dispersed randomly. The observed grouping was extremely unlikely to have happened by accident, as further supported by the z-score of 19.15. Furthermore, the statistical significance of the p-value of 0.000 indicated that there were unique regional trends in stunting throughout Sub-Saharan Africa (Fig 1).

Statistically significant clusters of high and low prevalence were identified by the hotspot analysis map, which displays the spatial distribution of stunting prevalence among children in Sub-Saharan Africa (Fig 2). Hotspots having 90%, 95%, or 99% confidence levels for stunting clustering were shown by regions that are highlighted in red. These hotspots, which showed regions with a significant burden of stunting, were primarily found in central and eastern Africa, including portions of South Sudan, Ethiopia, and the Democratic Republic of the Congo. On the other hand, cold spots were represented by regions in blue, which at comparable confidence levels indicated a noticeably low prevalence of stunting (Fig 2).

The majority of these chilly places were found in southern Africa, which includes South Africa, Namibia, and Botswana. The "Not Significant" (gray) areas highlighted the spatial heterogeneity of stunting in the area by indicating no discernible

**Table 2. Sociodemographic characteristics of study participants and their mothers, 2025.**

| Characteristics | Categories | Number | Percentage |
|---|---|---|---|
| Sex | Female | 86,630 | 49.62 |
| | Male | 87,956 | 50.38 |
| Residency | Urban | 45,914 | 26.30 |
| | Rural | 128,672 | 73.70 |
| Wealth index | Poorest | 49,332 | 28.26 |
| | Poorer | 37,881 | 21.70 |
| | Middle | 34,933 | 20.01 |
| | Richer | 29,499 | 16.90 |
| | Richest | 22,941 | 13.14 |
| Maternal education attainment | No education | 74,237 | 42.52 |
| | Incomplete primary | 38774 | 22.21 |
| | Complete primary | 17,237 | 9.87 |
| | Incomplete secondary | 29,101 | 16.67 |
| | Complete secondary | 10,065 | 5.77 |
| | Higher | 5,172 | 2.96 |

**Table 3. Proportion of stunting for eligible SSA countries, 2025.**

| Eligible countries | Stunting (breastfeeding Child) | | | | |
|---|---|---|---|---|---|
| | Yes | Prevalence | No | Prevalence | Total |
| Burkina Faso | 1,284 | 20.12 | 5,096 | 79.88 | 6,380 |
| Benin | 2,306 | 31.25 | 5,072 | 68.75 | 7,378 |
| Burundi | 4,784 | 53.68 | 4,128 | 46.32 | 8,912 |
| Congo DR | 4,786 | 42.40 | 6,500 | 57.6 | 11,286 |
| Cote;e Devore | 1,192 | 22.99 | 3,992 | 77.01 | 5,184 |
| Cameroon | 1,182 | 27.95 | 3,046 | 72.05 | 4,228 |
| Ethiopia | 4,180 | 35.54 | 7,580 | 64.46 | 11,760 |
| Gabon | 576 | 21.68 | 2,080 | 78.32 | 2,656 |
| Ghana | 938 | 18.53 | 4,124 | 81.47 | 5,062 |
| Gambia | 810 | 17.63 | 3,782 | 82.37 | 4,592 |
| Gunie | 1,244 | 30.44 | 2,842 | 69.56 | 4,086 |
| Kenya | 1,850 | 18.36 | 8,226 | 81.64 | 10,076 |
| Comoros | 337 | 28.22 | 857 | 71.78 | 1,194 |
| Liberia | 854 | 30.71 | 1,926 | 69.29 | 2,780 |
| Lesotho | 388 | 32.27 | 814 | 67.73 | 1,202 |
| Madagascar | 4,470 | 39.43 | 6,864 | 60.57 | 11,334 |
| Mali | 1,446 | 26.98 | 3,914 | 73.02 | 5,360 |
| Mauritania | 1,392 | 25.67 | 4,030 | 74.33 | 5,422 |
| Malawi | 1,864 | 34.07 | 3,606 | 65.93 | 5,470 |
| Mozambique | 1,310 | 32.42 | 2,730 | 67.58 | 4,040 |
| Nigeria | 4,332 | 35.35 | 7,922 | 64.65 | 12,254 |
| Niger | 679 | 38.15 | 1,101 | 61.85 | 1,780 |
| Namibia | 250 | 21.01 | 940 | 78.99 | 1,190 |
| Rwanda | 1,636 | 33.00 | 3,322 | 67 | 4,958 |
| Sierra Leone | 1,182 | 26.55 | 3,270 | 73.45 | 4,452 |
| Chad | 4,898 | 39.26 | 7,578 | 60.74 | 12,476 |
| Togo | 1,082 | 27.09 | 2,912 | 72.91 | 3,994 |
| Tanzania | 1,402 | 27.39 | 3,716 | 72.61 | 5,118 |
| Uganda | 1,334 | 26.60 | 3,680 | 73.4 | 5,014 |
| South Africa | 164 | 26.97 | 444 | 73.03 | 608 |
| Zimbabwe | 978 | 22.53 | 3,362 | 77.47 | 4,340 |
| **Total** | **55,130 (31.58%)** | | **119,456 (68.42%)** | | **174,586 (100%)** |

grouping trend. To address the high frequency of stunting, our analysis emphasizes the necessity of focused public health interventions in designated hotspots.

The spatial interpolation map showed a clear regional difference in the frequency of childhood stunting in Sub-Saharan Africa. Green-shaded regions with low stunting prevalence (0.00–0.33) were mostly found along the coast and in southern regions, such as South Africa, Namibia, and portions of Botswana. Yellow indicated areas of Tanzania, Zambia, and some West African regions with moderate prevalence (0.33–0.49). Orange and red-hued regions with high to extremely high prevalence (0.49–0.82) were found in central and eastern regions, including Ethiopia and the Democratic Republic of the Congo. Significant regional disparities were highlighted by these findings, emphasizing high-burden areas that urgently require focused interventions (Fig 3).

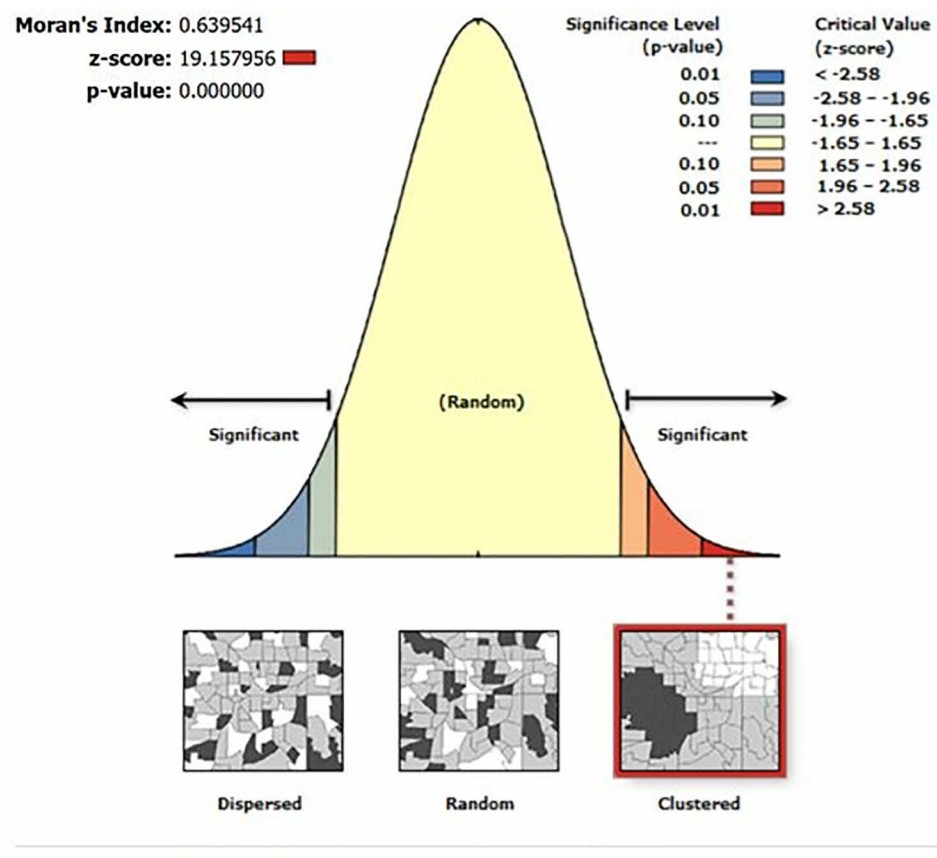

Given the z-score of 19.1579558976, there is a less than 1% likelihood that this clustered pattern could be the result of random chance.

**Fig 1. Autocorrelation for stunting among breast feeding children in SSA, 2025.**

## Discussion

In the current study, the proportion of stunting ranged between 17.63 and 53.68 percent. The study's results demonstrate the notable regional clustering of stunting among nursing infants in SSA. The high z-score (19.16), statistically significant p-value (<0.001), and Moran's Index value of 0.639541 suggested that the prevalence of stunting was not randomly distributed but rather showed clear regional trends.

Our finding agrees with related geographic disparities in stunting observed by Akombi et al. [30]. Although some countries have made progress, others continue to report high rates, indicating unequal and gradual growth. According to previous studies [31,32], the average stunting rate in SSA was 40%. The impact of regional elements on child nutritional outcomes in SSA is highlighted by this clustering, including socioeconomic inequality, cultural customs, access to healthcare, and environmental factors.

Significant clusters of high stunting prevalence were found by the hotspot analysis, demonstrating the existence of nutritional disparities both within and between SSA nations. These findings highlight the necessity of region-specific approaches to stunting prevention like previous studies done in African nations [33–36]. For instance, addressing underlying socioeconomic variables, boosting breastfeeding practices, improving food security, and strengthening mother and child healthcare facilities should be the top priorities of interventions in designated hotspots. Furthermore,

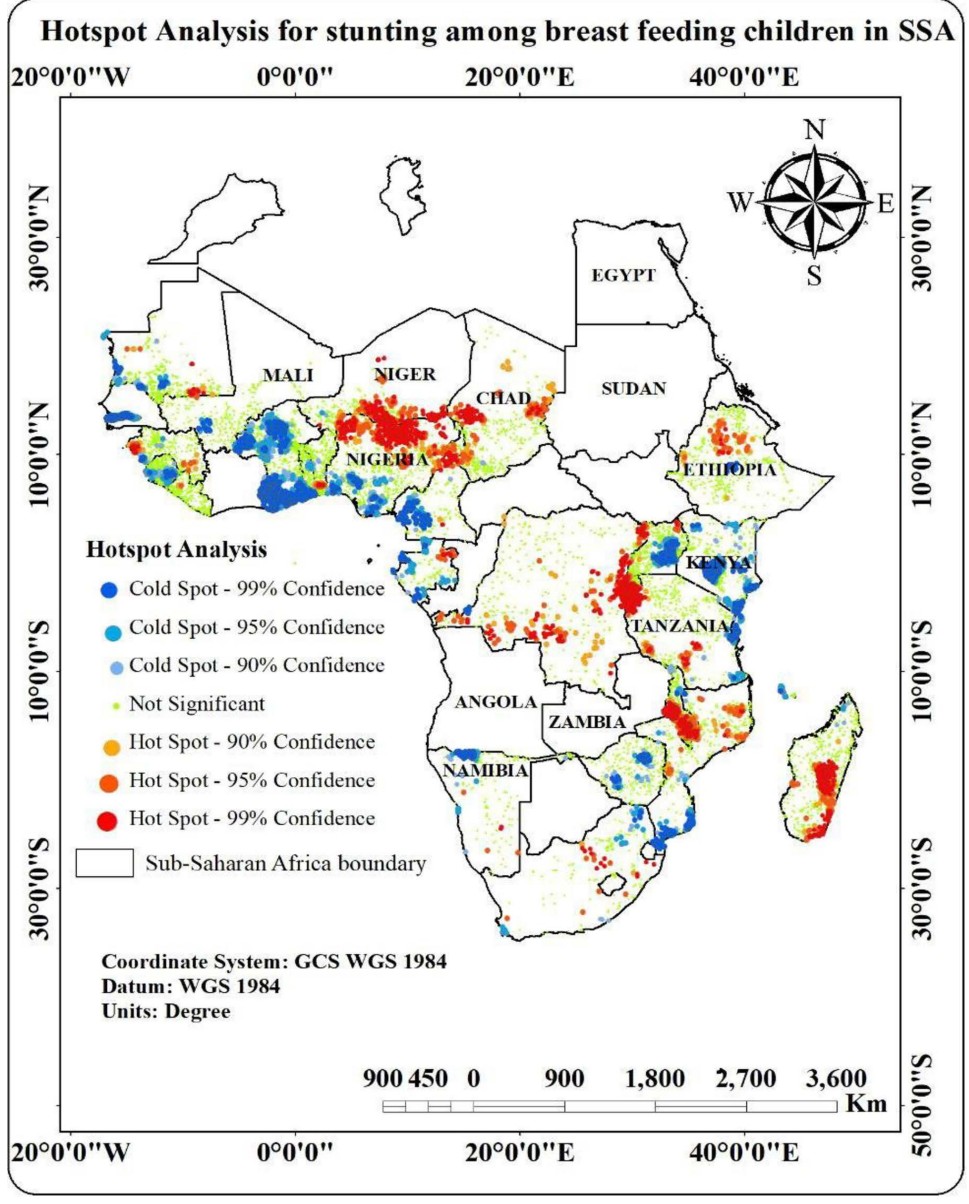

**Fig 2. Hotspot analysis for stunting among breast feeding children in SSA, 2025 (Africa boundary data accessed from** https://hub.arcgis.com/datasets/geoduck:africa-boundaries/about**).**

identifying cold spots offers chances to research effective interventions and duplicate them in regions with high frequency.

These findings were further supported by the spatial interpolation analysis, which showed that locations with a high prevalence of stunting were concentrated in particular places. Critical hotspots are indicated by these geographic trends, especially in regions of Central, Eastern, and Western Africa where stunting is still remarkably common. These results were consistent with earlier research indicating that some SSA regions are disproportionately affected by inadequate maternal nutrition, food instability, and restricted access to healthcare services [16,37,38]. On the other hand, regions

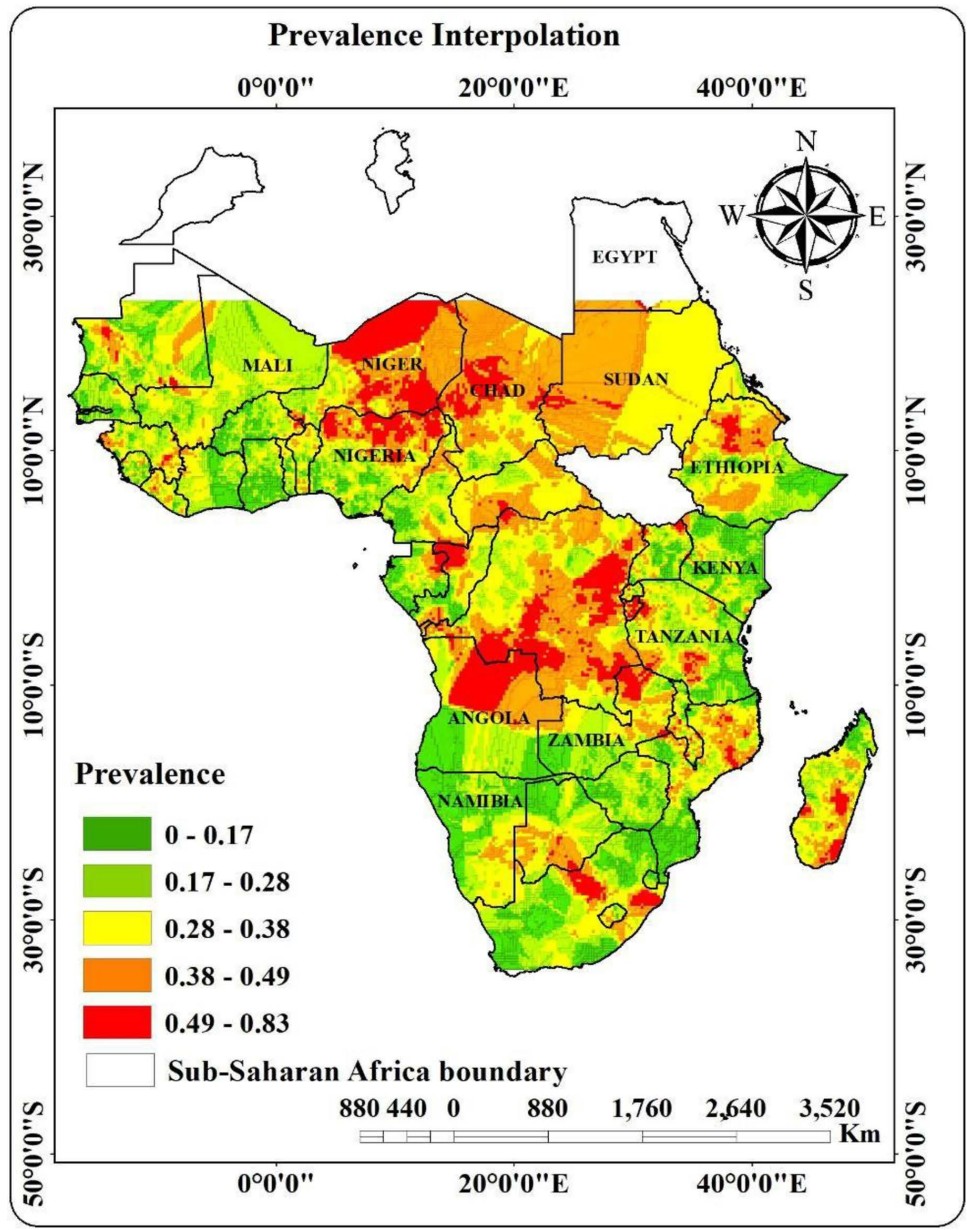

**Fig 3. Interpolation for stunting among breast feeding children in SSA, 2025 (Africa boundary data accessed from** https://hub.arcgis.com/datasets/geoduck:africa-boundaries/about**).**

known as "cold spots" that had lower rates of stunting might gain from increased food security, better maternal education, and easier access to healthcare.

Significant implications for focused interventions and public health policy in Sub-Saharan Africa result from the research's conclusions. Once health officials have identified regional hotspots of stunting, they can better allocate resources and prioritize populations that need them the most. Spatial insights can help guide the development of region-specific nutrition initiatives that target the causes of stunting. Interventions should focus on improving mother nutrition, breastfeeding practices, and supplemental feeding, for instance, in areas where stunting among breastfed children is rather prevalent.

Real-time decision-making and the long-term assessment of policy efficacy can also be facilitated by incorporating spatial analysis into national health surveillance systems. Last but not least, interventions that are geographically informed can help the region achieve sustainable development goals about nutrition and child well-being, improve child health outcomes, and lessen health disparities.

Despite this study identifying the cold and hotspot areas, it had limitations. Firstly, variations in data collecting among countries and recall bias may have an impact on DHS data. Inter-country comparability may be limited by variations in survey schedule and cultural customs. The real burden of stunting may also be underestimated if deceased children are excluded. Secondly, the factors for the difference in distribution haven't been assessed. Therefore, future studies should investigate the fundamental causes of these regional trends, such as environmental elements like agricultural production, climate variability, and access to water and sanitation. Additionally, incorporating geospatial analysis into national nutrition surveillance systems may improve the ability to identify individuals at risk and direct resources to the most underserved areas.

## Conclusion

Significant spatial clustering of stunting among breastfeeding children was found in this study throughout Sub-Saharan Africa, suggesting that the prevalence of stunting was localized in particular geographic areas rather than being dispersed randomly. Parts of Central, Eastern, and Western Africa have significant rates of stunting prevalence, as determined by Moran's Index, hotspot studies, and interpolation analyses. This highlighted the urgent need for region-specific treatments. These results emphasize the significance of implementing focused policies and initiatives to address regional causes of stunting, including poverty, limited access to healthcare, food insecurity, and cultural customs. In Sub-Saharan Africa, reducing the frequency of stunting and the long-term effects it causes can be achieved by using geospatial analytics to inform public health planning, resource allocation, and intervention tactics.

## Acknowledgments

The authors would like to thank the DHS program office for their permission to access the datasets and the GIS data of each country.

## Author contributions

**Conceptualization:** Bekahegn Girma, Azizur Rahman.

**Data curation:** Bekahegn Girma.

**Formal analysis:** Bekahegn Girma, Lealem Dibku Sasahu, Azizur Rahman.

**Funding acquisition:** Bekahegn Girma, Azizur Rahman.

**Investigation:** Bekahegn Girma, Azizur Rahman.

**Methodology:** Bekahegn Girma, Lealem Dibku Sasahu, Azizur Rahman.

**Project administration:** Bekahegn Girma, Azizur Rahman.

**Resources:** Bekahegn Girma.

**Software:** Bekahegn Girma, Lealem Dibku Sasahu.

**Supervision:** Bekahegn Girma, Lealem Dibku Sasahu, Azizur Rahman.

**Validation:** Bekahegn Girma, Lealem Dibku Sasahu, Azizur Rahman.

**Visualization:** Bekahegn Girma, Azizur Rahman.

**Writing – original draft:** Bekahegn Girma.

**Writing – review & editing:** Bekahegn Girma, Lealem Dibku Sasahu, Azizur Rahman.

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
