## [Decision Letter · Decision Letter 0]

Dear Dr. Girma, 

Thank you for submitting your manuscript to PLOS ONE. After careful consideration, we feel that it has merit but does not fully meet PLOS ONE’s publication criteria as it currently stands. Therefore, we invite you to submit a revised version of the manuscript that addresses the points raised during the review process.

We look forward to receiving your revised manuscript.

Kind regards,

Yibeltal Alemu Bekele, MpH

Academic Editor

PLOS ONE

Journal Requirements:

5. We note that Figure 2 and 3 in your submission contain map/satellite images which may be copyrighted. All PLOS content is published under the Creative Commons Attribution License (CC BY 4.0), which means that the manuscript, images, and Supporting Information files will be freely available online, and any third party is permitted to access, download, copy, distribute, and use these materials in any way, even commercially, with proper attribution. For these reasons, we cannot publish previously copyrighted maps or satellite images created using proprietary data, such as Google software (Google Maps, Street View, and Earth). For more information, see our copyright guidelines: http://journals.plos.org/plosone/s/licenses-and-copyright.

 a. You may seek permission from the original copyright holder of Figure 2 and 3 to publish the content specifically under the CC BY 4.0 license. 

6. Please ensure that you refer to Figure 2 and 3 in your text as, if accepted, production will need this reference to link the reader to the figure.

Reviewers' comments:

Reviewer's Responses to Questions

**Comments to the Author**

1. Is the manuscript technically sound, and do the data support the conclusions?

Reviewer #1: Yes

Reviewer #2: Yes

2. Has the statistical analysis been performed appropriately and rigorously?

Reviewer #1: Yes

Reviewer #2: Yes

3. Have the authors made all data underlying the findings in their manuscript fully available?

Reviewer #1: Yes

Reviewer #2: Yes

4. Is the manuscript presented in an intelligible fashion and written in standard English?

Reviewer #1: Yes

Reviewer #2: Yes

Reviewer #1: This manuscript presents valuable original research on the spatial distribution of stunting in SSA. With some minor revisions and additional details, it has the potential to make a significant contribution to the field of public health and nutrition in Sub-Saharan Africa.

Reviewer #2: Summary

The paper addresses an important issue. However, some areas require clarification and revision. The following comments highlight specific aspects that need revision:

Abstract:

• Please specify the spatial interpolation technique.

Methods:

• Study setting and period: The first sentence mentions that the current study was carried out in 48 countries, while the last line mentions that, due to some technical reasons, the study was carried out in 31 SSA countries. This inconsistency may cause confusion. Consider rephrasing for clarity.

• Population and sample: In the first line of the first paragraph, mention that the study focuses on children under five years of age. In the second paragraph, the fourth line after the full stop is unclear. Breastfeed part and the questioning of mothers of reproductive age seem combined. Consider separating these aspects for better clarity. The last paragraph of this section would be more effective if placed at the beginning to provide context for the study.

• Variables and data sources: Correct the abbreviation for Global Administrative Areas, to GADM.

• Mention the software used to create Figure 1.

• Provide the formula for calculating Moran’s I value.

Results:

• The first sentence repeats information already provided in the Methods section. Consider removing it to avoid redundancy.

• Introduce a subheading for the section discussing socio-demographic characteristics and present the associated findings in a separate paragraph before moving on to other results.

• Spatial distribution of stunting in SSA: Cite the relevant figure or table wherever appropriate. For example, in the second paragraph, the fourth and eighth lines, indicate the corresponding figure number.

Discussion:

Strengthen the discussion by comparing and contrasting the findings with previous studies. If possible, include a comparison of current estimates and patterns of stunting in SSA with data from an earlier period.

Figures and Tables:

• In Figure 2, provide coordinates and scale, indicate the north line, and add a heading for proper interpretation and analysis of the map.

• In figure and table captions, the year 2024 is mentioned as the reference year. Although, in Table 1, it is mentioned that the recent DHS data vary by country.

• In Tables 2 and 3, replace ‘frequency’ with ‘percentage/proportion’ for numerical values (Table 2) and for stunting prevalence across eligible countries (Table 3).

Recommended course of action: Request Revision

**Do you want your identity to be public for this peer review?** For information about this choice, including consent withdrawal, please see our Privacy Policy

Reviewer #1: **Yes: ** Luqman Adewale Abass

Reviewer #2: No

---

## [Author Response · Author response to Decision Letter 1]

25 Apr 2025

Title: Spatial distribution of stunting among breastfeeding children in Sub-Sahara Africa

Authors

Bekahegn Girma, Lealem Dibku Sasahu and Azizur Rahman

Dear Editor, thank you for giving us a chance to revise our manuscript entitled “Spatial distribution of stunting among breastfeeding children in Sub-Saharan Africa.” The reviewer’s comments are insightful and important to improve the quality of our manuscript. Based on the comments, we have made revisions and rewritten a point-by-point response. Finally, the clean version of the revised manuscript and the track changes are uploaded.

Reviewer 1

This manuscript presents valuable original research on the spatial distribution of stunting in SSA. With some minor revisions and additional details, it has the potential to make a significant contribution to the field of public health and nutrition in Sub-Saharan Africa.

Response: Dear reviewer, Thank you. We revised this manuscript accordingly based on your valuable comments.

Comments: Originality and Novelty: The study presents original research on the spatial distribution of stunting among breastfeeding children in Sub-Saharan Africa (SSA). While previous studies have examined stunting in specific countries, this research provides a comprehensive analysis across 31 SSA nations, filling a gap in the literature.

Response: Yes, you are right; the finding will help policy makers.

Comments: Methodology and Technical Standards: The researchers employed a robust methodology using Demographic and Health Survey (DHS) data from 31 SSA countries, encompassing 174,586 breastfeeding children across 15,398 clusters. The use of Geographic Information Systems (GIS) and spatial analytic methods, such as Getis-Ord Gi* and Global Moran's I, demonstrates a high technical standard in analyzing spatial patterns of stunting.

Response: Dear reviewer, thank you.

Comments: Data Analysis and Presentation: The statistical analysis appears to be conducted rigorously, with appropriate weighting to account for varying response rates and sampling probabilities. The use of spatial autocorrelation techniques and interpolation methods provides a comprehensive view of stunting distribution across SSA.

Response: Dear reviewer, thank you again.

Comments: Results and Conclusions: The study's findings are presented clearly, with stunting prevalence ranging from 17.63% to 53.68% across different SSA countries. The identification of hotspots in Central and Eastern Africa and cold spots in Southern Africa is well-supported by the data and analysis presented. The conclusions drawn from these results appear appropriate and align with the study's objectives.

Response: Dear reviewer, your idea is constructive and meaningful.

Comments: Clarity and Language: The manuscript is written in Standard English and is generally intelligible. However, there are some minor grammatical errors and awkward phrasings that could be improved for clarity.

Response: Extensive editing is done to avoid any grammatical errors and confusing paragraphs.

Comments: Ethical Considerations: While not explicitly stated in the provided passage, the use of DHS data typically adheres to ethical standards. However, the manuscript should include a statement on ethical approval and data usage permissions to ensure full compliance with research integrity standards.

Response: Thank you. Based on your suggestion, we revised the ethical consideration of our manuscript (page 12, line numbers 275-280).

Comments: Reporting Guidelines and Data Availability: The study appears to follow appropriate reporting guidelines for spatial analysis research. However, a statement on data availability should be included to meet community standards.

Response: The data availability section of this study is revised as you suggested (page 12, line numbers 282-284).

Comments: Literature Review: The background section could benefit from a more comprehensive review of recent literature on stunting in SSA, particularly studies using similar spatial analysis techniques.

Response: Dear reviewer, Thank you very much for your comments. We have tried to add comprehensive reviews conducted at the SSA level. However, there is no study conducted to assess the spatial distribution of stunting at the African level (page 3, line numbers 65-75).

Comments: Discussion: Expand on the implications of the findings for public health policy and interventions in SSA.

Response: The implication of this study is included in the discussion section in detail based on your suggestion (page 10, line numbers 236-247).

Comment: Limitations: Include a dedicated section discussing the study's limitations, such as potential biases in DHS data or challenges in cross-country comparisons

Response: As you recommended, an additional potential limitation of this study related to the nature of the data is included in the revised version of this manuscript (page 11, line numbers 151-256).

Reviewer 2

Comments: The paper addresses an important issue. However, some areas require clarification and revision. The following comments highlight specific aspects that need revision: Abstract: Please specify the spatial interpolation technique.

Response: Dear reviewer, thank you very much. The spatial interpolation technique (Kriging) is included in the abstract (page 2, line number 36).

Comments: Study setting and period: The first sentence mentions that the current study was carried out in 48 countries, while the last line mentions that, due to some technical reasons, the study was carried out in 31 SSA countries. This inconsistency may cause confusion. Consider rephrasing for clarity.

Response: The paragraph is paraphrased accordingly to enhance its clarity (page 4, line numbers 103-107).

Comments: Population and sample: In the first line of the first paragraph, mention that the study focuses on children under five years of age. In the second paragraph, the fourth line after the full stop is unclear. Breastfeeding and the questioning of mothers of reproductive age seem combined. Consider separating these aspects for better clarity. The last paragraph of this section would be more effective if placed at the beginning to provide context for the study.

Response: The above section of this manuscript is revised as you recommended, and the last paragraph is placed at the beginning of the paragraph (page 5, line numbers 109-119).

Comments: Variables and data sources: Correct the abbreviation for Global Administrative Areas to GADM. Mention the software used to create Figure 1. Provide the formula for calculating Moran’s I value.

Response: The abbreviation is corrected as well as the software for the autocorrelation is described in the clear version of this paper. Lastly, the Moran’s index formula is included (page 5-6, line numbers 128-153).

Comments: Results: The first sentence repeats information already provided in the Methods section. Consider removing it to avoid redundancy.

Response: It is removed.

Comment: Introduce a subheading for the section discussing socio-demographic characteristics and present the associated findings in a separate paragraph before moving on to other results.

Response: The subheading is inserted, and other variables are also described (page 7, line numbers 170-172).

Comments: Spatial distribution of stunting in SSA: Cite the relevant figure or table wherever appropriate. For example, in the second paragraph, the fourth and eighth lines indicate the corresponding figure number.

Response: the relevant figures are cited in the correct place in the manuscript (page 8, line numbers 180-193)

Comments: Discussion: Strengthen the discussion by comparing and contrasting the findings with previous studies. If possible, include a comparison of current estimates and patterns of stunting in SSA with data from an earlier period.

Response: Dear reviewer, Thank you for your constructive comments. Despite there being no similar study done to assess the spatial distribution, we used related studies and reports for comparison (page 9, line numbers 209-213).

Comments: In Figure 2, provide coordinates and scale, indicate the north line, and add a heading for proper interpretation and analysis of the map.

Response: The coordinate, scale, and heading of Figure 2 are inserted.

Comment: In figure and table captions, the year 2024 is mentioned as the reference year.

Response: All the captions are checked, and the year is changed to 2025.

Comments: Although, in Table 1, it is mentioned that the recent DHS data vary by country.

Response: Dear reviewer, Thank you for your concern. However, there is no common and fixed year for each country. Therefore, we used the recent DHS data of each country (within 10 years).

Comments: In Tables 2 and 3, replace ‘frequency’ with ‘percentage/proportion’ for numerical values (Table 2) and for stunting prevalence across eligible countries (Table 3).

Response: A revision is done accordingly as you advised.

Thank you!

---

## [Decision Letter · Decision Letter 1]

Spatial distribution of stunting among breast feeding children in Sub-Sahara Africa

PONE-D-25-03856R1

Dear Dr. Bekahegn Girma,

We’re pleased to inform you that your manuscript has been judged scientifically suitable for publication and will be formally accepted for publication once it meets all outstanding technical requirements.

Kind regards,

Yibeltal Alemu Bekele, MpH

Academic Editor

PLOS ONE

---

## [Editor Report · Acceptance letter]

PONE-D-25-03856R1

PLOS ONE

Dear Dr. Girma,

I'm pleased to inform you that your manuscript has been deemed suitable for publication in PLOS ONE. Congratulations! Your manuscript is now being handed over to our production team.

Kind regards,

on behalf of

Mr. Yibeltal Alemu Bekele

Academic Editor

PLOS ONE